# Differential Spatial Gene and Protein Expression Associated with Recurrence Following Chemoradiation for Localized Anal Squamous Cell Cancer

**DOI:** 10.3390/cancers15061701

**Published:** 2023-03-10

**Authors:** Sharia Hernandez, Prajnan Das, Emma B. Holliday, Li Shen, Wei Lu, Benny Johnson, Craig A. Messick, Cullen M. Taniguchi, John Skibber, Ethan B. Ludmir, Y. Nancy You, Grace Li Smith, Brian Bednarski, Larisa Kostousov, Eugene J. Koay, Bruce D. Minsky, Matthew Tillman, Shaelynn Portier, Cathy Eng, Albert C. Koong, George J. Chang, Wai Chin Foo, Jing Wang, Luisa Solis Soto, Van K. Morris

**Affiliations:** 1Translational Molecular Pathology, The University of Texas—MD Anderson Cancer Center, Houston, TX 77030, USA; sdhernandez@mdanderson.org (S.H.);; 2Gastrointestinal Radiation Oncology, The University of Texas—MD Anderson Cancer Center, Houston, TX 77030, USA; 3Bioinformatics, The University of Texas—MD Anderson Cancer Center, Houston, TX 77030, USA; 4Gastrointestinal Medical Oncology, The University of Texas—MD Anderson Cancer Center, Houston, TX 77030, USA; 5Colon and Rectal Surgery, The University of Texas—MD Anderson Cancer Center, Houston, TX 77030, USA; 6Vanderbilt-Ingram Cancer Center, Nashville, TN 37232, USA; 7Pathology, The University of Texas—MD Anderson Cancer Center, Houston, TX 77030, USA

**Keywords:** squamous anal cell carcinoma, chemoradiation, recurrence, protein profiling, digital spatial profiling

## Abstract

**Simple Summary:**

While anti-PD1 antibodies have demonstrated efficacy in some patients with metastatic anal cancer, these agents have no proven benefit for those with localized disease treated with chemoradiation. Difficulty procuring fresh tumor tissue required for RNA and protein expression analysis has limited extensive molecular profiling for this rare cancer. Our team utilized a novel digital spatial profiling technology on pretreatment anal cancer specimens to identify biomarkers associated with recurrence after chemoradiation. We observed that recurrent tumors had higher baseline expression of immune checkpoint biomarkers, higher MAPK signaling activation and higher PI3K/Akt signaling activation. These findings provide a rationale that supports future clinical trials with immunotherapy that seek to improve survival beyond chemoradiation for patients with localized squamous cell cancer of the anus.

**Abstract:**

The identification of transcriptomic and protein biomarkers prognosticating recurrence risk after chemoradiation of localized squamous cell carcinoma of the anus (SCCA) has been limited by a lack of available fresh tissue at initial presentation. We analyzed archival FFPE SCCA specimens from pretreatment biopsies prior to chemoradiation for protein and RNA biomarkers from patients with localized SCCA who recurred (N = 23) and who did not recur (N = 25). Tumor cells and the tumor microenvironment (TME) were analyzed separately to identify biomarkers with significantly different expression between the recurrent and non-recurrent groups. Recurrent patients had higher mean protein expression of FoxP3, MAPK-activation markers (BRAF, p38-MAPK) and PI3K/Akt activation (phospho-Akt) within the tumor regions. The TME was characterized by the higher protein expression of immune checkpoint biomarkers such as PD-1, OX40L and LAG3. For patients with recurrent SCCA, the higher mean protein expression of fibronectin was observed in the tumor and TME compartments. No significant differences in RNA expression were observed. The higher baseline expression of immune checkpoint biomarkers, together with markers of MAPK and PI3K/Akt signaling, are associated with recurrence following chemoradiation for patients with localized SCCA. These data provide a rationale towards the application of immune-based therapeutic strategies to improve curative-intent outcomes beyond conventional therapies for patients with SCCA.

## 1. Introduction

Squamous cell cancer of the anal canal is caused by prior infection with human papillomavirus (HPV) in more than 90% of cases [1]. The availability of a preventative HPV vaccine is expected to reduce drastically the incidence of HPV-associated cancers such as anal cancer in the United States [2,3,4]. However, the annual incidence of anal cancer continues to increase in the United States [5], with more than 9000 diagnoses expected in 2023 [6]. Most patients present with localized disease [5], where concurrent chemoradiation is the standard curative intent treatment [7,8]. Although most patients experience excellent clinical outcomes with this multimodality approach, those who have persistent disease or recur may undergo abdominoperineal resection with permanent end colostomy [9,10].

The classification of patients with anal cancer at high risk for recurrence after chemoradiation has been limited to clinicopathologic characteristics, including male gender and more advanced lymph node positive or bulky and advanced T4 primary tumor [11,12,13,14]. Genomic annotations have yielded limited benefit given the low tumor mutation burden and rarity of actionable mutations characteristic of anal cancer [15,16,17]. Alterations in genes activating PI3K/Akt signaling are common in patients with non-metastatic [15] and metastatic [16] anal cancer alike and represent a targetable driver for the oncogenesis of these cancers. Mutations in oncogenes such as *KRAS* and *BRAF* are prevalent in adenocarcinomas of the adjacent colon and rectum [18] and promote the development of these cancers. However, therapies targeting these mutations have historically not been utilized as a treatment for anal cancer. To date, immune checkpoint blockade agents have demonstrated anti-tumor efficacy for patients with metastatic anal cancer [19,20,21]. In these series, translational analyses of small subsets of patients have suggested that PD-L1 expression has been linked to improved treatment efficacy and survival outcomes. These findings have yet to be validated for patients with non-metastatic anal cancer and could support a rationale towards the development of novel immunotherapy approaches for patients with curable disease. To our knowledge, no molecular biomarker analysis has reported utility for the risk stratification of patients with anal cancer prior to the initiation of chemoradiation. Feasibility for such characterization of RNA and protein characterization of anal cancer has proven difficult historically due to a lack of availability of the needed fresh tissue in a rare cancer for which most patients can be cured without surgery.

Recent advances in sequencing methodologies have demonstrated ability to quantify transcriptomic and proteomic expression spatially within the tumor microenvironment using archival, formalin-fixed, paraffin-embedded (FFPE) tumor tissue [22]. In order to address this unmet need for identifying molecular alterations prognostic for survival in this setting, we conducted protein and gene expression profiling analysis that compared pretreatment anal cancer specimens, in the tumor and the tumor microenvironment areas, between patients with newly diagnosed anal cancer who achieved cure versus those who recurred after definitive chemoradiation.

## 2. Materials and Methods

### 2.1. Identification of Patient Samples

The University of Texas—MD Anderson Cancer Center medical record was retrospectively reviewed to identify patients who were diagnosed with nonmetastatic squamous cell carcinoma of the anal canal between 2014–2020 for participation on an IRB-approved protocol for analysis of archival tissue. Patients with archival FFPE tumor blocks of untreated squamous cell cancer of the anus/anal canal obtained at the time of initial presentation (i.e., prior to any treatment with radiation or chemotherapy) were included. These patients were classified based on their recurrent or non-recurrent oncologic status following completion of chemoradiation. Because there were more patients in our database who did not recur after chemoradiation, we selected available cases to balance the demographic and clinical characteristics with the recurrent cohort. Here, clinical and pathologic data were collected from medical records for these patients. Comparisons between non-recurrent and recurrent groups were performed using unpaired t-tests (SPSS) for continuous variables. A Fisher’s exact test was used to assess associations for clinical and pathologic features unique to the non-recurrent and recurrent groups.

### 2.2. Digital Spatial Profiling

For each patient, hematoxylin-eosin (H&E) stained slides of tumor specimens from a freshly sectioned FFPE tumor block were reviewed with optic microscopy by a pathologist to perform quality control (QC) based on the histologic appearance of the samples. A minimum cutoff of 200 cells was required per biological segment (tumor or tumor microenvironment) to select adequate regions for further evaluation with NanoString GeoMx Digital Spatial Profiling (DSP) for both RNA and protein assay [23]. Samples with extensive necrosis or hemorrhage were not included. Following confirmation of QC, two consecutive 5 µm thick sections—one for the protein assay and the other for RNA assay—were stained with the semi-automated GeoMx DSP standard protocol [24] using the Leica Bond Rx autostainer system (Leica Biosystem) to profile the tumor and tumor microenvironment. For identification of tumor and cells of the tumor microenvironment, the RNA assay sections were stained with pancytokeratin (panCK), CD45, SYTO 13 (GeoMx Solid Tumor TME Morphology Kit, Cat# 121300301) and CD68 (Clone KP-1, A594, catalog number: SC-20060, Santa Cruz, concentration 1:2000 µg/mL), and the protein assay sections were stained with pancytokeratin (panCK), CD45 and SYTO 13 (GeoMx Solid Tumor TME Morphology Kit, Cat# 121300301) and CD3e (Clone UMAB54, AF647, catalog number: UM500048, Origene, concentration 0.25 µg/mL).

For protein profiling, we used 73 immune related biomarkers with the following panels (Appendix A): GeoMx Immune Cell Profiling (Cat# GMX-PROCO-NCT-HICP-12), GeoMx IO Drug Target (Cat# GMX-PROMOD-NCT-HIODT-12), GeoMx Immune Activation Status (Cat# GMX-PROMOD-NCT-HIAS-12), Immune Cell Typing (Cat# GMX-PROMOD-NCT-HICT-12), GeoMx Cell Death (Cat# GMX-PROMOD-NCT-HCD-12), GeoMx MAPK Signaling (Cat# GMX-PROMOD-NCT-HMAPK-12), GeoMx PI3K/AKT Signaling (Cat# GMX-PROMOD-NCT-HPI3K-12). In the other sectioned tumor slide, we performed RNA profiling with the GeoMx Immune immune-related RNA targets (Appendix A). Following completion of the scanning in the GeoMx DSP device, multiplex immunofluorescence image slides were visualized adjusting channel thresholds for each fluorophore.

### 2.3. Selection of Regions of Interest (ROI)

Selection of ROIs was performed after pathology evaluation of sequential sections of H&E staining to secure the minimum quantity of cells per Area of Illumination (AOI) per assay: 200 cells for RNA and 20 cells for protein. Rectangle selection tool was applied to select up to 5 ROIs of 660 × 785 μm on each case, based on the expression of CD45 (immune-enriched areas). Each ROI was segmented into biological compartments, or area of illumination (AOI), distinguished as “tumor” (panCK-positive) or “tumor microenvironment (TME)” (panCK-negative). Areas selected for protein assay were matched for overlapping analysis with the RNA assay (Figure 1). Segmented AOIs were illuminated individually via ultraviolet light on the GeoMx DSP device, photocleaving the oligonucleotides tags conjugated with antibodies present within each AOI. Released tags were quantitated in an nCounter and counts were mapped back to tissue location, yielding a spatially resolved digital profile of analyte abundance. Digital counts were normalized using housekeepers.

For DSP protein expression evaluation, a total of 435 AOIs were selected (220 from tumor areas and 215 segmented from TME). After initial QC, 2 TME areas and 1 tumor area were flagged as “high positive control normalization” and were excluded from the study, leaving 219 tumor areas (114 non-recurrent, 105 recurrent) and 213 tumor microenvironment areas (111 non-recurrent, 102 recurrent) that passed final QC for protein expression. After data normalization, median DSP counts for both tumor and TME compartments were aligned when comparing recurrent with non-recurrent tumor samples.

For DSP RNA expression evaluation, a total of 443 AOIs (225 from tumor areas and 218 from TME areas) were selected. After initial QC, a total of 220 tumor (non-recurrent 116, recurrent 104) and 218 tumor microenvironment (non-recurrent 109, recurrent 199) areas were considered for final analysis of RNA expression. After data normalization, median DSP counts for both tumor and TME compartments were aligned when comparing recurrent with non-recurrent samples.

### 2.4. Data Analysis

Comparisons for expression for individual gene and protein biomarker mean expression were measured as a log2 fold-change (FC) for recurrent: non-recurrent groups. A linear model with two variables was used to fit to the protein and RNA data so that the recurrence effect (recurrent vs. non-recurrent) could be adjusted for differences between the patients. The analysis was carried out using limma package in R (version 3.6.3), with an empirical Bayes method to moderate the standard errors of the estimated log-fold. This results in more stable inference and improved power especially for experiments with small numbers of samples. To adjust for multiple testing, Benjamini and Hochberg (1995) method was used to control false discovery rate (FDR).

## 3. Results

### 3.1. Demographics

Of the 50 patients with anal cancer (25 recurrent and 25 non-recurrent) initially identified, there were 48 (23 recurrent and 25 non-recurrent) whose baseline/untreated anal cancer specimens satisfied the histology quality control necessary for digital spatial profiling. The demographic characteristics of these patients are detailed in Table 1. Patients in the recurrent group were more likely to be treated with a fluoropyrimidine/mitomycin C doublet for their chemoradiation than the non-recurrent group (57% vs. 20%, *p* = 0.01); all other patients received a fluoropyrimidine/platinum doublet for their definitive treatment. No other differences in underlying clinical and pathologic features were noted between the non-recurrent and recurrent groups. The mean age at diagnosis was 59.5 years (standard deviation (SD), 10.0) for non-recurrent patients and 56.6 years (SD 9.0) for recurrent patients (*p* = 0.30). Most patients in both cohorts (60% non-recurrent and 74% recurrent) had stage III disease at the time of the initial presentation of anal cancer, with no differences between the two groups in terms of distribution of disease stage at initial presentation (*p* = 0.36). Human papillomavirus was detected in 95% of evaluable cases for NR and R groups. Most tumors were poorly differentiated (50% non-recurrent and 57% recurrent groups, *p* = 0.56). For the recurrent patients, there were 10 (43%) who recurred locally and 13 (57%) who developed distant metastases after definitive therapy.

### 3.2. Differential Protein Expression between Recurrent and Non-Recurrent Patients

Within the tumor compartments, significant differential protein expression was detected in baseline tumors between recurrent and non-recurrent patients (Figure 2A). Relative to the non-recurrent group, recurrent patient samples had statistically significant higher mean expression of fibronectin (FC 2.59, *p* = 0.002), T cell regulatory associated protein Foxp3 (FC 1.77, *p* = 0.005), T cell activation Granzyme B (FC 1.43, *p* = 0.02), proliferation and cycle cell regulation Phospho−p38 MAPK (T180/Y182) (FC 1.64, *p* = 0.04) and BRAF (FC 1.52, *p* = 0.045) (Table 2). Within the tumor compartment, we did not find significant differences in biomarkers of T-cell infiltration (CD3 or CD8), myeloid or immune checkpoint biomarker expression or biomarkers associated with cell death or PI3K/AKT signaling pathways.

However, within the tumor microenvironment, significant differential expression for immune biomarkers was notable between recurrent and non-recurrent patients (Figure 2B). Here, biomarkers associated with immune T cell activation such as Granzyme B (FC 1.56, *p* = 0.008) and OX40 L (FC 1.70, *p* = 0.03) were higher in recurrent patients (Table 3). In addition, immune checkpoint biomarkers associated with inhibitory functions of T cell activation such as PD-L2 (FC 1.74, *p* = 0.04), PD-1 (FC 1.45, *p* = 0.03) and LAG3 (FC 1.61, *p* = 0.04) and with regulatory T cells (e.g., FoxP3 (FC 1.86, *p* = 0.004)) were also significantly higher in specimens of patients with recurrent tumors. Interestingly, analysis of MAP kinase signaling pathway expression in the TME showed the higher expression of BRAF (FC 1.75, *p* = 0.005), Phospho−p38 MAPK (T180/Y182) (FC 1.76, *p* = 0.006), Phospho−p90 RSK (T359/S363) (FC 1.52, *p* = 0.01), Phospho−MEK1 (S217/S221) (FC 1.69, *p* = 0.006) in the recurrent patients. In the PI3K/AKT signaling pathway, higher expression of Phospho−GSK3A (S21)/Phospho−GSK3B (S9) (FC 1.58, *p* = 0.01) and Phospho−AKT1 (S473) (FC 1.63, *p* = 0.01) was notable in the recurrent cohort relative to the non-recurrent patients. Similar to the tumor component, higher mean fibronectin was also noted in the recurrent group within the tumor microenvironment (FC 3.41, *p* = 0.0002).

### 3.3. RNA Expression in Recurrent and Non-Recurrent Patients

No statistically significant RNA expression was found in gene expression associated with T cell or myeloid cell infiltration, immune checkpoints, immune activation and cytokine pathways, between recurrent and non-recurrent patients in tumoral or tumor microenvironment areas (Figure 3).

## 4. Discussion

Here, we present, to our knowledge, the first spatial comparison for gene and protein expression between patients with locoregional squamous cell carcinoma of the anal canal who were and were not cured by subsequent chemoradiation. Historically, such analyses have been limited by the absence of fresh tissue in pretreatment samples of this rare malignancy. The utilization of a novel sequencing methodology with available FFPE tumor tissue allowed us not only to perform RNA and protein profiling but also to characterize differences in biomarker expression between tumor cells and the adjacent tumor microenvironment that associate with recurrences in patients with anal cancer.

Fibronectin protein expression was significantly elevated in both tumor cells and the tumor microenvironment alike for patients with anal cancer who recurred after chemoradiation relative to the cohort of patients who were cured. High fibronectin expression has been associated with inferior survival for patients with other HPV-associated cancers such as cervical cancer [25] and head/neck cancer [26]. In vitro, models of HPV-associated cancers have linked fibronectin overexpression to increased activation of the focal adhesion kinase signaling pathway, which promotes cancer cell migration [27] and the polarization of anti-inflammatory M2 macrophages [28] that may promote tumor progression [29]. Consistent with these findings for other HPV-associated malignancies, our data here provide further support for fibronectin as an unfavorable prognostic biomarker for anal cancer.

The differential protein expression of clinically actionable immune biomarkers within the anal cancer tumor microenvironment was noted between patients who recurred versus who did not recur following chemoradiation. For example, the increased expression of FoxP3, associated with the regulation of regulatory T cells [30] and immune tolerance to cancer cells within the tumor microenvironment [31,32], occurred within both the tumor cell and tumor microenvironment compartments for patients experiencing disease recurrence in our study. Uniquely for the tumor microenvironment, the higher expression of immune checkpoint biomarkers such as PD-1, PD-L2, LAG-3 and OX40L was observed in the clinically unfavorable cohort of recurrent anal cancer patients.

For patients with unresectable and/or metastatic squamous cell carcinoma of the anal canal, treatment with immune checkpoint blockade therapies has demonstrated modest clinical activity, with response rates ranging between 10 and 24% [19,20,21,33,34]. In small post hoc correlative analyses for these clinical trials, the increased expression of PD-L1 has been associated with a greater likelihood for response to therapy [19,21]. To date, no data have been reported detailing outcomes of immunotherapy for patients with localized anal cancer undergoing curative-intent therapy, whether before or after chemoradiation. Given the availability of therapeutic agents targeting these multiple immune checkpoints, our data here provide an initial rationale that warrants further validation for testing (combination) immunotherapy approaches in patients with locoregional anal cancer, especially in patients who are at higher risk for recurrence following chemoradiation.

We also observed the higher expression of biomarkers associated with radioresistance in pretreatment specimens of the TME of patients with recurrent anal cancer. Specifically, higher phospho-Akt expression (associated with PI3K/mTOR signaling activation and radioresistance) [35,36,37], higher BCL-6 (associated with anti-apoptotic activity in response to DNA damage-induced stress) [38,39] and higher phospho-GSKβ (associated with the activation of oncogenic Wnt/β-catenin, MAPK, PI3K/mTOR and Notch signaling activity) [40,41] were measured at higher levels in this cohort relative to their counterparts who were eventually cured by chemoradiation. Based on these findings, it is possible that tumors in the prognostically unfavorable anal cancer group who recurred were predisposed to overcome induced stress on the DNA by both cytotoxic chemotherapy and radiotherapy in order to account for the inability to achieve a curative outcome by multimodality treatment. Future trials should consider the analysis of these biomarkers in evaluating response to chemoradiation, with the eventual goal of the application of matched targeted therapies as treatment to overcome de novo radiotherapy and/or chemotherapy resistance.

Of interest, the expression of proteins associated with MAPK signaling activation was associated with anal cancer recurrence following chemoradiation. Prior trials in this setting have demonstrated no improvement in clinical outcomes yet added treatment-related mortality with the anti-EGFR antibody cetuximab in improving outcomes for patients with locoregional anal cancer treated with chemoradiation [42,43,44,45]. This EGFR target lies proximally within this oncogenic MAPK signaling pathway, which has been linked to adaptive resistance to radiation in other cancer types [46,47,48] besides anal cancer. In addition, it is notable that the increased expression of upstream activators of ERK (BRAF and MEK1) and of p38 were observed in the tumor microenvironment component of patients with recurrent anal cancer. MAPK signaling has been reported to increase the transformation of primary fibroblasts into cancer-associated fibroblasts within the stromal compartment in melanoma [49]. Therefore, it is possible that MAPK activation may link with the promotion of immune suppression biomarkers such as FoxP3 characteristic of patients with the clinically unfavorable recurrent tumors.

Notably, there was no overlap between RNA and protein biomarkers that were observed in our findings. It is possible that epigenetic modulation described for HPV-associated cancers [50] such as anal cancer may have affected post-translational protein expression, which may account for this discrepancy. Nonetheless, we opted to focus our analysis here on the protein biomarkers that distinguished the groups of anal cancer that were either cured or recurred after chemoradiation. That multiple regions of interest were selected and evaluated within each tumor using this validated methodology may ensure that intertumoral heterogeneity was accounted for in analysis for each patient. While we acknowledge that the results obtained in this study could be strengthened by other orthogonal assays such as single immunohistochemistry (IHC) for the confirmation of protein expression, this was not possible due to the exhaustion of small-volume core biopsies that prevented any additional analyses. We also recognize that the performance of this work at a single high-volume academic referral center may not reflect the diversity of patients who are diagnosed with anal cancer, specifically in reference to a low number of available patients living with human immunodeficiency virus included here. We emphasize the critical need to conduct similar studies representative of the different subsets of patients diagnosed with anal cancer. It is our hope that the historical limitations of quantifying RNA and protein expression may be overcome with technologies such as that described here in order to improve scientific advances for all patients with this orphan malignancy. If these prognostic biomarkers are validated in larger series, their clinical applicability could justify the further study of novel radiation sensitizers and/or treatment strategy investigation chemotherapy/immunotherapy induction treatments prior to chemoradiation. In addition, with the forthcoming EA2165 study evaluating the addition of nivolumab after chemoradiation for patients with high-risk anal cancer pending readout, the potential to add immunotherapy to patients identified at high risk for recurring after chemoradiation represents another possible therapeutic application for consideration.

## 5. Conclusions

In summary, we present here novel biomarkers associated with de novo resistance to chemoradiation for patients with localized anal cancer. While chemoradiation has been the treatment in this setting for more than half a century, clinical practice has historically relied upon demographic and clinical features to identify patients at “high risk” for ensuing treatment failure. The advent of new methods to quantify RNA and protein expression using small amounts of FFPE tumor specimens addresses a knowledge gap for identifying prognostic biomarkers associated with survival outcomes, even for rare malignancies such as anal cancer. Ultimately, these findings may be applied towards the analysis of pretreatment specimens for the development of novel therapeutic approaches that potentiate the activity of chemoradiation, with the ultimate goal of curing more patients with anal cancer.

## Figures and Tables

**Figure 1 cancers-15-01701-f001:**
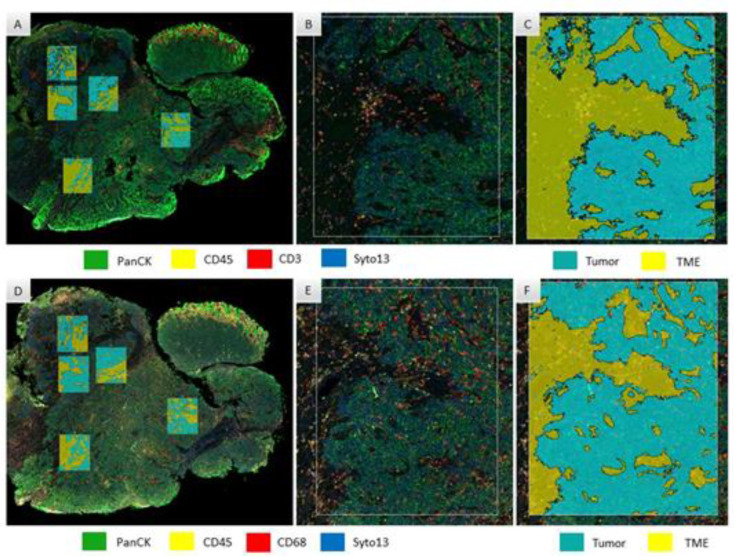
Region of interest selection and segmentation for protein (**A**–**C**) and RNA (**D**–**F**) digital spatial profiler assay in an anal carcinoma sample. All the areas selected for protein assay were matched for RNA assay, here seen with low magnification in (**A**,**D**), respectively; and at high magnification for one of the regions of interest in (**B**,**E**), respectively (colors for the immunofluorescence morphology staining are seen in rectangles on the bottom of these panels). (**C**,**F**) show tissue segmentation in the same high magnification area, for tumor and tumor microenvironment compartments (colors indicating mark-up areas for tumor and TME are seen on the bottom of these panels).

**Figure 2 cancers-15-01701-f002:**
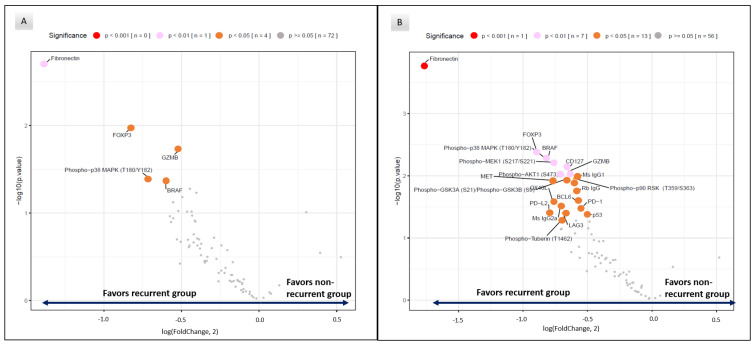
Volcano plots of differentially expressed protein biomarkers for tumor (**A**) and tumor microenvironment (**B**) compartments in recurrent vs. non-recurrent samples.

**Figure 3 cancers-15-01701-f003:**
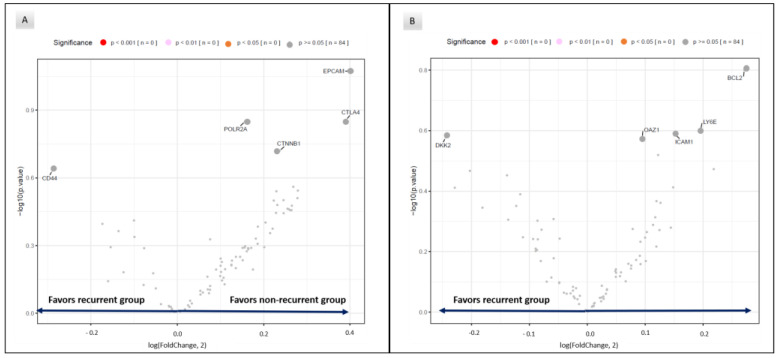
Volcano plots of RNA biomarkers for tumor (**A**) and tumor microenvironment (**B**) compartments in recurrent vs. non-recurrent samples.

**Table 1 cancers-15-01701-t001:** Patient Demographics.

	Non-Recurrent (N = 25)	Recurrent (N = 23)	*p*-Value *
Age, years (SD) **	59.5 (10.0)	56.6 (9.0)	0.30
Gender (%)			0.10
Female	18 (72)	21 (91)	
Male	7 (28)	2 (9)	
Ethnicity (%) ***			0.22
African-American	0 (0)	2 (9)	
Caucasian	22 (88)	20 (87)	
Hispanic	3 (12)	1 (4)	
Stage at Diagnosis (%)			0.36
I	1 (4)	0 (0)	
II	9 (36)	5 (22)	
III	15 (60)	17 (74)	
IV	0 (0)	1 (4)	
HPV status (%)			10.0
Positive	18 (72)	18 (78)	
Negative	1 (4)	1 (4)	
Not available	6 (24)	4 (17)	
HIV Status (%)			0.46
Negative	25 (100)	22 (96)	
Positive	0 (0)	1 (4)	
Differentiation (%)			0.56
Well	1 (4)	0 (0)	
Moderately	10 (40)	10 (43)	
Poorly	11 (44)	13 (57)	
Not available	3 (12)	0 (0)	
Coexisting autoimmune disease (%)			0.36
Absent	22 (88)	22 (94)	
Present	3 (12)	1 (4)	
Chemotherapy with radiation			0.01
Fluoropyrimidine +			
mitomycin C	5 (20)	13 (57)	
Fluoropyrimidine +			
cisplatin	20 (80)	10 (4)	

* *p*-value < 0.05 was considered statistically significant. ** (SD): Standard deviation values for age values inside brackets. *** (%): Percentages values for ethnicity, stage at diagnosis, HPV status, differentiation and coexisting autoimmune disease, inside brackets.

**Table 2 cancers-15-01701-t002:** Fold-change (log2) differential mean protein expression between recurrent and non-recurrent patients within tumor compartment.

Biomarker	Expression Fold Change (Recurrent: Non-Recurrent)	*p*-Value *
Fibronectin	2.59	0.002
FoxP3	1.77	0.005
GZMB	1.43	0.02
Phospho−p38 MAPK (T180/Y182)	1.64	0.04
BRAF	1.52	0.045

* *p*-value < 0.05 was considered statistically significant.

**Table 3 cancers-15-01701-t003:** Fold-change in differential mean protein expression between recurrent and non-recurrent patients within tumor microenvironment compartment.

Biomarker	Fold Change (Recurrent: Non-Recurrent)	*p*-Value *
Fibronectin	3.41	0.0002
FOXP3	1.86	0.004
GZMB	1.56	0.008
Phospho−p38 MAPK (T180/Y182)	1.76	0.006
BRAF	1.75	0.005
CD127	1.58	0.008
PD−L2	1.74	0.04
OX40L	1.70	0.03
PD-1	1.45	0.03
LAG3	1.61	0.04
BCL6	1.50	0.03
p53	1.42	0.04
MET	1.72	0.01
Phospho−GSK3A (S21)/Phospho−GSK3B (S9)	1.58	0.01
Phospho−MEK1 (S217/S221)	1.69	0.006
Phospho−p90 RSK (T359/S363)	1.52	0.01
Phospho−AKT1 (S473)	1.63	0.01

* *p*-value < 0.05 was considered statistically significant.

## Data Availability

The data presented in this study is available in this article and Appendix A. Additional information is available from the corresponding author on reasonable request.

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
