# Peer review of "Differential Spatial Gene and Protein Expression Associated with Recurrence Following Chemoradiation for Localized Anal Squamous Cell Cancer"

_cancers, 2023, doi:10.3390/cancers15061701_

Round 1
Reviewer 1 Report
In this manuscript, the authors examined stored tissue and changes in gene and protein expression using a multiplex approach and record the differences between patients that were either cured or not cured using chemoradiation. For those not cured, one might expect the presence of sustained activation of cell signalling pathway(s) associated with proliferation and this was quantified by the authors. The manuscript, therefore, provides an interesting insight into the pathways that might need to be targeted for those patients that do not fully respond to chemoradiation treatment. The methods and results are well described and for such a discrete and finite study that is well presented, I have specific recommendations for revision as the paper is suitable for publication. The authors could provide an independent methodology follow-up to substantiate their results such as by using immunohistochemistry to localize and quantify the relative changes in say one or two of the proteins that exhibited extensive changes between the cured vs non-cured groups. However, this would be a recommendation but not an absolute requirement for publication. The authors have produced an interesting and focused study; hence, I would just suggest the following before publication:
1. A further check to pick up the typographical and grammatical errors.
2. There are a number of protein changes documented in the abstract but the inclusion of the word significant would confirm to the reader that these are not trends but are statistically validated changes.
3. An adjustment to Figure 2 and 3 since the colour changes associated with levels of significance is too difficult to read on the panels.
Author Response
- The authors could provide an independent methodology follow-up to substantiate their results such as by using immunohistochemistry to localize and quantify the relative changes in say one or two of the proteins that exhibited extensive changes between the cured vs non-cured groups. However, this would be a recommendation but not an absolute requirement for publication
We acknowledge and appreciate the reviewer’s comment about incorporating an independent methodology to validate the results of this study. However, since the tissue used for our research reported here came from (core) needle biopsies, all tissue was exhausted following initial clinical pathology use and subsequent digital spatial profiling detailed here. We do not have additional tissue available to perform such needed work in an adequate number of matched patients. We have clarified this important point from the reviewer as a limitation to interpretation of our results at sentence 5 of paragraph 7 of the Discussion section.
- A further check to pick up the typographical and grammatical errors.
We appreciate the reviewers’ feedback, and have reviewed and corrected typographical and grammatical errors in the updated (tracked changes) version of this manuscript. In making these modifications, we did not alter any findings or message of our manuscript.
- There are a number of protein changes documented in the abstract but the inclusion of the word significant would confirm to the reader that these are not trends but are statistically validated changes.
We have included the word significant throughout the updated version of the manuscript to clarify statistically significant results, when appropriate.
- An adjustment to Figure 2 and 3 since the colour changes associated with levels of significance is too difficult to read on the panels.
We thank the reviewer for the suggestion, and we have edited Figures 2 and 3 to visualize better significant changes.
Reviewer 2 Report
In the manuscript entitled "Differential spatial gene and protein expression associated with recurrence following chemoradiation for localized anal squamous cell cancer" the authors proposed as they stated the first study regarding a spatial comparison of gene and protein expression for patients suffering from anal canal squamous cell carcinoma.
The study brings new information regarding the possible biomarkers specific for chemoradiation resistance in patients suffering from anal cancer. The method proposed can become useful for other type of cancers and can help in development of new therapeutic approaches.
However, several issues need to be addressed in the manuscript.
The authors should describe briefly in the introduction section some of the biomarkers presented in the results section.
For table 1, the authors should insert a footnote which explains that the values in bracket are the SD.
Also the significance level should be clearly mentioned as well for all 3 tables.
Author Response
- The authors should describe briefly in the introduction section some of the biomarkers presented in the results section.
We appreciate the reviewer’s suggestion here and accordingly have incorporated in the updated manuscript a brief introduction into biomarkers associated with PI3K/Akt, MAPK, and immunce checkpoint activation for the landscape of anal cancer into the Introduction section.
- For table 1, the authors should insert a footnote which explains that the values in bracket are the SD. Also the significance level should be clearly mentioned as well for all 3 tables.
Per the reviewer’s suggestion we have inserted the footnote to better explain values in the table including Standard Deviation and p values.